# Programmed Death-Ligand 1-Positive Squamous Cell Carcinoma Spontaneously Regressed after Percutaneous Needle Biopsy

**DOI:** 10.3390/medicina59030631

**Published:** 2023-03-22

**Authors:** Masayuki Sasahara, Hiroki Takahashi, Takashi Ohchi, Naohiro Nomura, Kentaro Kodama, Kimiyuki Ikeda, Hirotaka Nishikiori, Kenzo Okamoto, Hirofumi Chiba

**Affiliations:** 1Department of Respirology, Sokujin-kai Kitahiroshima Hospital, Kitahiroshima 061-1121, Japan; 2Department of Respiratory Medicine and Allergology, Sapporo Medical University School of Medicine, Sapporo 060-8556, Japan; 3Department of Pathology, Hokkaido Chuo Rosai Hospital, Iwamizawa 068-0004, Japan

**Keywords:** lung cancer, pneumonia, spontaneous regression, percutaneous needle biopsy, programmed death-ligand 1

## Abstract

Spontaneous lung cancer regression is a very rare course of disease. A 60-year-old male patient was admitted to our hospital with pneumonia and a 19 mm-sized nodule shadow in the S4 of the left lung on chest computed tomography (CT). A percutaneous needle biopsy was performed, and a diagnosis of programmed death-ligand 1-positive squamous cell lung carcinoma was made based on pathological findings. The patient was followed up with imaging because the lesion has reduced in size on chest CT. We report the possibility that cellular immune mechanisms triggered by needle biopsy contributed to spontaneous regression.

## 1. Introduction

Spontaneous regression (SR) of cancer is proposed as partial or complete lesion disappearance after no treatment or treatment considered ineffective and lasting at least 1 month. SR in malignant lymphoma, hepatocellular carcinoma, and lung cancer has been reported in a relatively small number of cases. Spontaneous lung cancer regression is a very rare course of disease, and patients with lung cancer are usually treated with surgery, chemotherapy and radiation therapy, etc., depending on the histological types, progression and general condition. An immune checkpoint inhibitor should be considered in patients with lung cancer with a high programmed death-ligand 1 (PD-L1) tumor proportion score (TPS) (>50%) expression to be treated with chemotherapy. Here, we report a case of squamous cell lung carcinoma diagnosed by percutaneous needle biopsy with high PD-L1 TPS expression (>50%), which spontaneously regressed without treatment.

## 2. Case Report

A 60-year-old male patient with symptoms of cough, sputum, and shortness of breath visited our hospital. The patient had a pacemaker implanted for complete atrioventricular block, was on medication for chronic heart failure, and smoked 20 cigarettes per day for 42 years. He was diagnosed with malignant lymphoma and treated at other hospital 20 years ago, with no findings suggestive of recurrence. His vital signs revealed no fever upon examination, but transcutaneous arterial oxygen saturation (SpO2) was 94% in room air, which was slightly low. Chest auscultation revealed no obvious heart murmur but coarse crackles on the left side. Blood tests showed a high white blood cell count of 8500 /µL (83.0% of neutrophils). Tumor markers were mildly elevated at the carcinoembryonic antigen (CEA) of 6.3 ng/mL, but other markers tested were within the standard values of cytokeratin 19 fragment (CYFRA) at 1.6 ng/mL and pro-gastrin releasing peptide (ProGRP) at 71.8 pg/mL. Chest computed tomography (CT) showed infiltration shadow of the lower lobe of the left lung and a nodule with irregular margins of 19 mm length in the S4 of the left lung (Figure 1). No obvious metastasis to other organs or lymph nodes was observed.

The patient was admitted to the hospital with a pneumonia diagnosis, and antimicrobial therapy was started on ceftriaxone (CTRX) at 2 g/day. Steroid therapy was not administered. The patient was discharged from the hospital after 14 days of antimicrobial therapy for a decreased inflammatory response on blood analysis and disappearance of infiltrative shadows on chest imaging although the causative bacterium could not be identified. After discharged from the hospital, there were no changes in his living environment, including smoking, dietary changes, or new medications or supplements. Laboratory tests and imaging studies on admission showed no findings suggestive of recurrent pneumonia. The patient was readmitted to the hospital 2 weeks after discharge, and a percutaneous needle biopsy was performed to confirm the diagnosis of an S4 nodule shadow in the left lung. The S4 nodule shadow in the left lung was similar on chest CT at biopsy. The patient was discharged from the hospital in good general condition after biopsy. Pathological examination revealed tumor cell proliferation by hematoxylin and eosin staining, and additional immunostaining was positive for p40, p63, and cytokeratin (CK) 5/6 and negative for thyroid transcription factor-1 (TTF-1) and Napsin A. Based on these results, the patient was diagnosed with squamous cell lung carcinoma (cT1bN0M0 stage ⅠA2 according to TNM eighth edition). Further pathological findings revealed no infiltration of NK cells, CD8-positive T cells and CD4-positive T cells while the degeneration of the tissue in the immediate vicinity of the tumor lesion and proliferation of CD68-positive macrophages (Figure 2). Additionally, tumors showed high PD-L1 TPS expression (>50%) (Figure 3).

The retrospective review of change on the chest CT showed that the lesion in S4 of the left lung obviously reduced 35 days after biopsy. After careful follow-up without therapeutic intervention, the tumor continued to shrink until 20 months after the biopsy, and no obvious metastasis to other organs was observed (Figure 4). Additionally, natural killer (NK) cell activity in peripheral brood 12 months after the biopsy was 16% (upper reference limit 40%). Tumor markers at 12 months after biopsy were CEA at 5.0 ng/mL and CYFRA at 1.6 ng/mL, which were not elevated. The patient continues to be observed for findings that may suggest re-enlargement of the lesion. The patient is followed up with periodic laboratory tests and imaging studies for findings suggestive of recurrence.

## 3. Discussion

SR of cancer is proposed as partial or complete lesion disappearance after no treatment or treatment considered ineffective and lasting at least 1 month [1]. The SR is extremely rare with approximately 1 in 60,000 to 100,000 patients with malignant tumors [2], which is higher (approximately 1 in 12,000) in Japan [3]. SR in malignant lymphoma, hepatocellular carcinoma, and lung cancer have been reported in a relatively small number of cases. A review of the literature by Challis and Stam from 1966 to 1987 also determined that of a total of 504 SR patients, only 25 (5%) had primary lung or bronchial carcinoma [4].

Zhang’s review of PubMed searches for 30 years from 1988 to January 2018 reported only 14 cases of primary lung cancer with SR in which pathological examination could explain complete or partial disappearance of cancer in the patient’s tissues [5]. Of the 14 cases, pathological samples were collected by percutaneous lung biopsy in four, transbronchial biopsy in five, thoracoscopy in one, sputa cytology in one, and unknown method in two. Recently, cases of lung cancer in which SR were observed after endobronchial ultrasound-guided transbronchial needle aspiration were reported [6,7]. Based on the literature from 1988 to 2019, 18 cases of lung cancer were compared by pathological type, with three of adenocarcinoma, eight of squamous cell carcinoma, none of small cell carcinoma, three of large cell carcinoma, and four of poorly differentiated carcinoma [8]. On the other hand, in a literature review in Japan, the frequencies are adenocarcinoma (45%), squamous cell carcinoma (20%), small cell carcinoma (20%), and large cell carcinoma (5%), which is almost the same as the incidence of cancer [3].

SR of malignant tumor is often considered sudden and triggered by external factors [9]. Mechanisms of SR have been extensively proposed, ranging from immune mediation, tumor inhibition by cytokines or growth factors, hormonal influence, elimination of carcinogenesis, tumor necrosis, angiogenesis inhibition, apoptosis, epigenetic mechanisms to induction of differentiation [9,10,11,12]. Regarding cancer immunity, it is well known that cytotoxic T cells and NK cells play a central role [13,14]. However, there are only a few case reports suggesting the involvement of these immune cells in SR cases, such as the report by Iwakami et al. stating that CD8-positive T cells may have played an important role in cancer cell cytotoxicity [15]. There are also reports of immunostaining for the percentage of CD4-positive T cells and CD8-positive T cells within tumors that have SR. Nogimori et al. reported a higher percentage of CD8-positive T cells within the tumor [16], while Ito et al. reported that CD4-positive T cells and CD8-positive T cells within the tumor were comparable, rather than fewer CD8-positive T cells at the tumor margin [17]. Since we could not confirm the infiltration of NK cells, CD8-positive T cells and CD4-positive T cells in the biopsy specimen obtained in our case, we have not obtained findings that support the above mechanism. 

On the other hand, a case with SR suggesting the involvement of prominent macrophage infiltration into the tumor tissue have been reported by Kawasaki et al. [18]. In their patient, lung mass lesion surgically resected after SR showed no viable tumor cells and the replacement by coagulative necrosis with prominent infiltration of macrophages. Similarly, in our case, both tumor tissue and necrotic areas were markedly infiltrated with CD68-positive macrophages, many of which expressed PD-L1. PD-L1 is a transmembrane protein that is expressed on tumor cells and binds to PD-1 on the surface of T cells to suppress tumor immunity [19]. PD-L1-expressing macrophages are commonly recognized as tumor-associated macrophages (TAMs), and the presence of TAMs promotes tumor growth [20]. However, in our case, the tumor disappeared after biopsy despite the presence of macrophages presumed TAMs, indicating the discrepancy between histopathological findings and subsequent clinical course. Although we do not have objective evidence to fully explain this paradoxical phenomenon, we presume that the mechanical stimulation induced by biopsy suppressed the interaction of PD-1 and PD-L1 via an immunological mechanism that mimics immune checkpoint inhibitors, resulting in the shrinkage of tumor.

The act of biopsy results in vascular injury and hemorrhage within the tumor microenvironment, releasing vasoactive substances from damaged tissue and platelets and increasing capillary permeability. As a result, leukocytes and plasma emerge into the tumor microenvironment. This early response to wound healing involves helper T cell type 1 response, which releases interferon-γ and interleukin-1 to promote the differentiation and migration of macrophages of the M1 phenotype [21] We hypothesize that through these processes, macrophages within the tumor microenvironment replaced the TAMs (which exhibits M2-like traits in malignant tumors) with the M1 phenotype and restored the ability of CD8-positive T cells to target the SR. However, in order for such phenotype conversion to result in SR, which occurs very rarely, it is necessary to sustain the conversion. In addition, recently, PI3Kγ has attracted attention as a key molecule of TAM, and it has been shown that inhibition of PI3Kγ reduces its own immunosuppressive function and promotes T cell cytotoxicity [22]. Therefore, it is possible that mechanical stimulation by biopsy had some effect on PI3Kγ expression.

As of October 2022, PubMed searches have detected only one report referring to the relationship with PD-L1 expression in cases in which spontaneous regression occurred without surgery, chemotherapy, or radiotherapy [8]. On the other hand, since the selection of anti-PD-1 antibody drugs considering expression not only in tumor cells but also in stromal cells is being carried out in clinical settings [23], changes in TAM related to spontaneous disappearance have to be focused on. Based on the above background, the relationship between PD-L1 expression on tumor cells, in the future, we hope that the relationship between PD-L1 expression on tumor cells, local infiltration of TAMs, and PD-1 expression on T cells will be investigated in detail in cases showing SR associated with interventional invasion. There have also been reports of changes in pathological findings on biopsies performed before spontaneous regression and after tumor re-enlargement in cases in which CD4-positive and CD8-positive T cells were thought to be associated with spontaneous regression [24]. If the lesions re-enlarge in the patient, comparison of histopathologic findings from biopsy or surgery may lead to an analysis of these associations. Furthermore, there are cases of spontaneous regression without mechanical invasion triggered by changes in the living environment or supplements, and a more detailed study of the mechanism may be possible if these cases can be compared.

## 4. Conclusions

We report a case of squamous cell lung carcinoma diagnosed by percutaneous needle biopsy with high PD-L1 TPS expression (>50%), which spontaneously regressed. The patient did not receive standard cancer treatments, including chemotherapy, nor did he make any changes in his living environment, diet, or start taking new medications or supplements. Since there are cases of SR, the possibility of lung cancer should be considered even if the nodule shadow shows shrinkage on chest imaging. We presume that the mechanical stimulation induced by biopsy suppressed the interaction of PD-1 and PD-L1 via an immunological mechanism that mimics immune checkpoint inhibitors, resulting in the shrinkage of tumor. We also hypothesize that macrophages within the tumor microenvironment replaced the TAMs (which exhibits M2-like traits in malignant tumors) with the M1 phenotype and restored the ability of CD8-positive T cells to target the SR. The mechanism of SR assumed by the pathological findings in our case is limited because it is based on observation of only a small portion obtained by needle biopsy. Although our case has progressed without re-enlargement of the lesion, some cases have been reported to show re-enlargement of the lesion after spontaneous regression [25], and careful long-term observation is considered necessary.

## Figures and Tables

**Figure 1 medicina-59-00631-f001:**
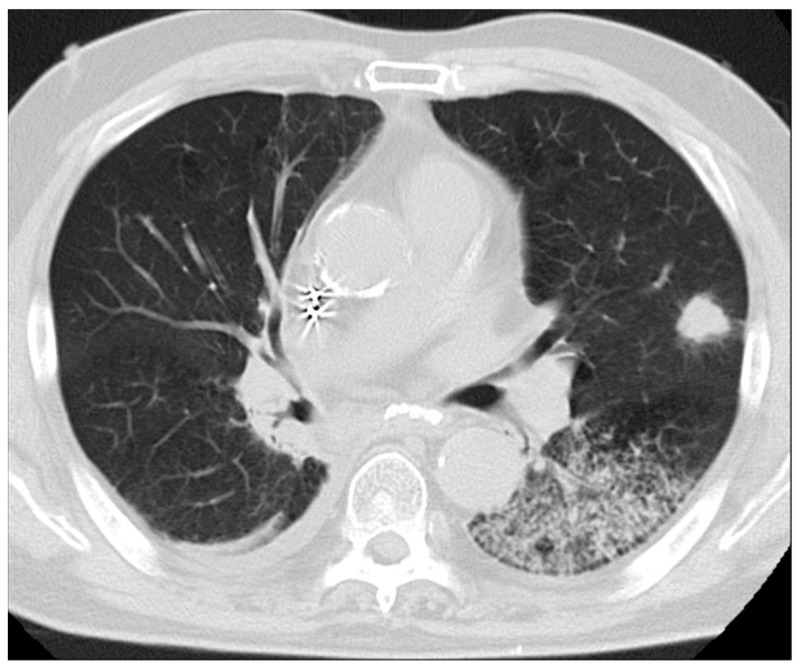
At the time of admission, chest CT showed infiltration shadow in the lower lobe of the left lung and a nodule with irregular margins of 19 mm length in the S4 of the left lung.

**Figure 2 medicina-59-00631-f002:**
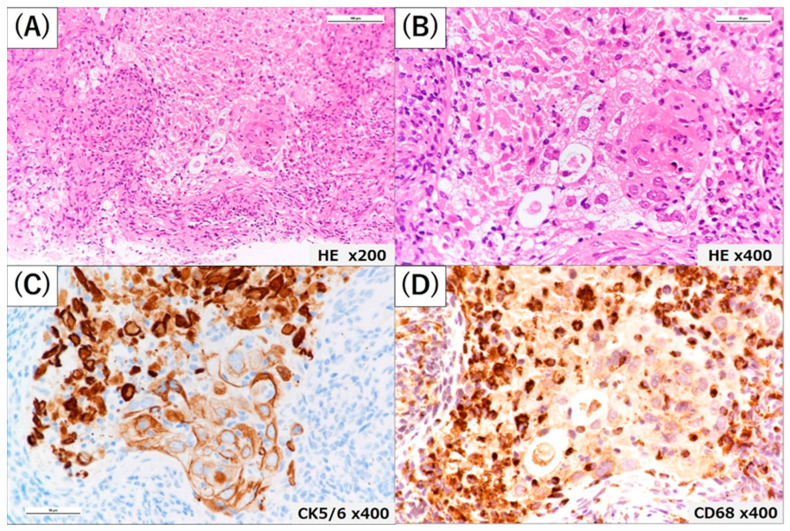
The histological examination of a percutaneous needle biopsy. At low power magnification (Hematoxylin and Eosin staining, 100×), the tumor grows in an irregular geographic pattern consisting of polygonal cells with brightly pink cytoplasm, and hyperchromatic and angular nuclei, and an area of necrosis in its upper can be seen (**A**). The pink cytoplasm with distinct cell borders and immature keratin pearls are seen at high magnification (HE 400×) (**B**). Cells positive for CK5/6 by immunohistochemistry are shown not only in the tumor but also in the necrotic area (400×) (**C**). CD68-positive cells (macrophages) by immunohistochemistry are shown more abundant in the necrotic area than in the tumor and its stroma (400×) (**D**).

**Figure 3 medicina-59-00631-f003:**
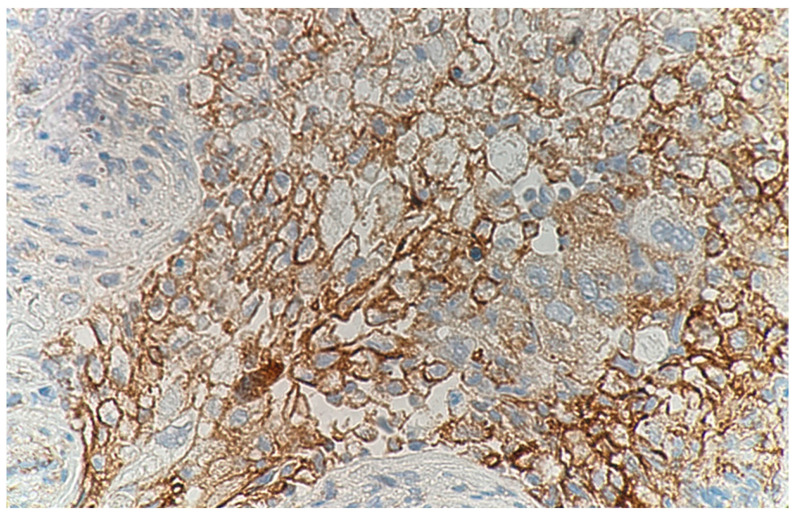
The immunohistochemistry using anti-PD-L1 clone 22C3 that targets the extracellular domain of PD-L1 (400×). Cells positive for PD-L1 are shown not only in the tumor but also in the necrotic area. PD-L1: programmed death-ligand 1.

**Figure 4 medicina-59-00631-f004:**
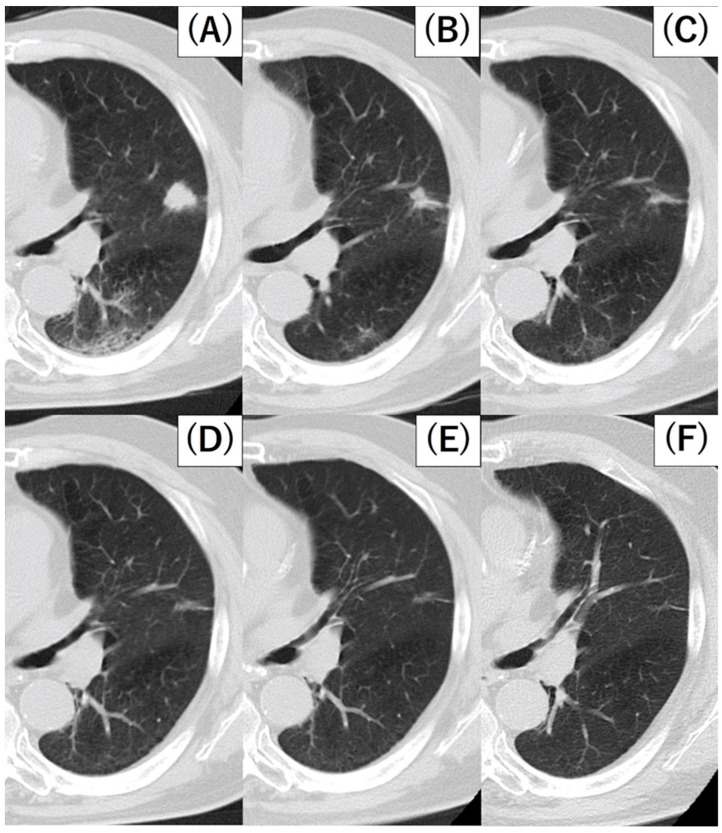
Chest CT images before biopsy (**A**), 35 days after biopsy (**B**), 67 days after biopsy (**C**), 118 days after biopsy (**D**), 12 months after biopsy (**E**), 20 months after biopsy (**F**). The S4 nodule in the left lung continued to shrink after biopsy without any anti-cancer therapy.

## Data Availability

Data supporting the study findings are available from the corresponding author upon reasonable request.

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
