# Peer review of "Programmed Death-Ligand 1-Positive Squamous Cell Carcinoma Spontaneously Regressed after Percutaneous Needle Biopsy"

_medicina, 2023, doi:10.3390/medicina59030631_

Round 1

Reviewer 1 Report

In this Case report the Authors describe a spontaneous regression of a squamous cell lung cancer with PD-L1 expression after CT-guided percutaneous needle biopsy. This event has been reported only in few previous reports, most of which without an analysis of PD-L1 expression, and the manuscript could therefore be of interest. Th Authors should address some minor points:

It should be reported if any steroid therapy was administered along with the antimicrobial therapy

Since, as the Authors report, previous reports have shown in some cases re-growth of the lesion after temporary regression, a follow-up longer than 20 months is required to draw definite conclusions.

Reviewer 2 Report

This is a rare and interesting case report of spontaneously regressed after percutaneous needle biopsy of programmed death-ligand 1-positive squamous cell carcinoma in the lung. This case report would be may provide some useful information on the spontaneously regressed of NSCLC (squamous cell carcinoma). I have a few minor comments.

1. Has the patient had a history of malignancy? Please describe in detail.

2. Page 1, line 41-42. “And no 41 obvious metastasis to other organs or lymph nodes was observed.”

: Did you perform PET/CT in this patient for nodule which located in the S4 of the left lung? If you performed PET/CT, please add the image of PET/CT.

3. Authors presume that the mechanical stimulation induced by biopsy suppressed the interaction of PD-1 and PD-L1 via an immunological mechanism that mimics immune checkpoint inhibitors, resulting in the shrinkage of tumor.

: How do you think about the shrinkage of tumor mass without biopsy in other case reports? What kind of immune response can explain this case?
